# Induction and Suspension Culture of *Panax japonicus* Callus Tissue for the Production of Secondary Metabolic Active Substances

**DOI:** 10.3390/plants13172480

**Published:** 2024-09-04

**Authors:** Siqin Lv, Fan Ding, Shaopeng Zhang, Alexander M. Nosov, Andery V. Kitashov, Ling Yang

**Affiliations:** 1Department of Biology, Shenzhen MSU-BIT University, Shenzhen 518172, China; dingfan0110@yandex.ru (F.D.); biology@kitashov.ru (A.V.K.); 2College of Forestry, Beijing Forestry University, Beijing 100091, China; 3State Key Laboratory of Tree Genetics and Breeding, Northeast Forestry University, Harbin 150040, China; lsq990124@163.com; 4National Key Laboratory of Plant Molecular Genetics, CAS Center for Excellence in Molecular Plant Sciences (CEMPS), Institute of Plant Physiology and Ecology (SIPPE), Chinese Academy of Sciences (CAS), Shanghai 200032, China; 5National Research and Development Center for SE-RICH Agricultural Products Processing, School of Life Sciences and Technology, Wuhan Polytechnic University, Wuhan 430023, China; shaopeng@whpu.edu.cn; 6Department of Plant Physiology, Biological Faculty, Lomonosov Moscow State University, Moscow 119991, Russia; al_nosov@mail.ru; 7Department of Cell Biology, Institute of Plant Physiology K.A. Timiryazev, Russian Academy of Sciences, Moscow 127276, Russia

**Keywords:** *Panax japonicus*, callus tissue, culture time, suspension culture, “total saponins” triterpenod glycosides, ginsenosides, accumulation

## Abstract

Using *Panax japonicus* as research material, callus induction and culture were carried out, and high-yielding cell lines were screened to establish a suspension culture system that promotes callus growth and the accumulation of the “total saponins” (total content of triterpenoid glycosides or ginsenosides). Using the root as an explant, the medium for callus induction and proliferation was optimized by adjusting culture conditions (initial inoculation amount, carbon source, shaking speed, hormone concentration, culture time) and a high-yielding cell line with efficient proliferation and high total saponins content was screened out. The conditions of suspension culture were refined to find out the most suitable conditions for the suspension culture of callus, and finally, the suspension culture system was established. We found that the lowest (5%) contamination rate was achieved by disinfecting the fresh roots with 75% alcohol for 60 s, followed by soaking in 10% NaClO for 15 min. The highest induction rate (88.17%) of callus was obtained using the medium MS + 16.11 μmol·L^−1^ NAA + 13.32 μmol·L^−1^ 6-BA + 30.0 g·L^−1^ sucrose + 7.5 g·L^−1^ agar. The callus was loose when the callus subcultured on the proliferation medium (MS + 5.37 μmol·L^−1^ NAA + 13.32 μmol·L^−1^ 6-BA + 30.0 g·L^−1^ sucrose + 3.8 g·L^−1^ gellan gum) for 21 days. The callus growth was cultured in a liquid growth medium (MS + 5.37 μmol·L^−1^ NAA + 13.32 μmol·L^−1^ 6-BA + 30.0 g·L^−1^ sucrose) with an initial inoculation amount of 40 g·L^−1^, a shaking speed of 110 r/min and darkness. Cell growth was fastest with a culture period of 21 days. We replaced the growth medium with the production medium (MS + 5.37 μmol·L^−1^ NAA + 13.32 μmol·L^−1^ 6-BA + 30.0 g·L^−1^ glucose) for maximum accumulation of total saponins. [Conclusion] A callus induction and suspension culture system for the root of *P. japonicus* was established. In this way, we can promote the accumulation of total saponins in callus cells and provide a basis for large-scale cell culture and industrial production of medicinal total saponins.

## 1. Introduction

*Panax japonicus* C. A. Mey, a perennial herb known as the ‘king of herbs’, is a valuable medicinal plant in China. Its primary medicinal constituents are total saponins [1], which have anti-tumor, anti-diabetic, anti-aging and anti-inflammatory properties, among others [2,3,4,5,6]. However, wild *P. japonicus* resources are scarce. Traditional seed and rhizome propagation methods are insufficient for large-scale cultivation due to the long natural growth cycle and low seedling rate, placing plants at risk of extinction [7]. Plant tissue culture and cell suspension culture are two techniques that can provide alternative methods for rapid cultivation. Plant tissue culture technology is unaffected by external environmental factors. It can enhance the reproduction rate of seedlings and produce secondary metabolites by artificially controlling the growth conditions [8], which can conserve the resources of *P. japonicus* to some extent. Zhu et al. [9] induced regenerated plants with tender stems and rhizomes but were unable to transplant them. Fukuoka Shōmi [10] et al. successfully induced regenerated plants using flower buds, stems, leaves and rhizomes of *P. japonicus* as explants. Liu [11] developed a suitable culture scheme for callus induction in *P. japonicus*; Luo [12] also improved the survival rate of tissue-cultured plants.

Cell suspension culture technology enhances the extraction rate of secondary metabolites and allows for the extraction of various secondary metabolites from suspended cells, significantly increasing their utilization rate. Zhang [13] et al. established the suspension culture system of *P. ginseng*, and Li [14] et al. optimized the system to improve the content of triterpenoid glycosides. Glagoleva [15] et al. investigated the growth and biosynthesis of 20-year-old suspension cells of *P. japonicus* by shake flask culture. Kochkin [16] also investigated the accumulation of ginsenosides in the process of *P. japonicus* cell culture. Research on callus induction and the suspension culture of *P. japonicus* in China is still in its early stages, requiring further systematic research. In this work, *P. japonicus* root was used as an explant to establish a callus induction and suspension culture system, aiming to provide a foundation for large-scale cell culture and industrial production of medicinal biologically active substances. Additionally, this research could pave the way for innovative germplasm technologies such as cell fusion, doubling and protoplast preparation of single cells and cell groups from suspension culture.

## 2. Results

### 2.1. Healing Callus Induction in Panax japonicus Roots

#### 2.1.1. Disinfection Method of *Panax japonicus* Root

Various disinfection methods (Appendix A) indicate that at a NaClO concentration of 3%, the contamination rate of the roots gradually decreased as the disinfection time increased. When the disinfection time was higher than 15 min, the contamination rate gradually increased, and the survival rate gradually decreased. A similar trend was observed at NaClO concentrations of 5% and 10%. At a 10% NaClO concentration, the lowest fouling rate was 5%, and no mortality was observed. To summarize, the optimal disinfection scheme 10 (Appendix A) for the roots, as determined in this study, involves disinfection with 75% alcohol for 60 s, followed by disinfection with 10% NaClO for 15 min.

#### 2.1.2. Screening of Induction Scheme for Root Callus of *Panax japonicus*

Comparing the mean k1, k2, k3 and k4 values (Appendix A) for the individual levels revealed that the optimal combination of NAA and 6-BA was I3 + II3 (treatment 11), i.e., the concentration of NAA was 16.11 μmol·L^−1^ and the concentration of 6-BA was 13.32 μmol·L^−1^. This treatment resulted in the highest callus induction rate for the roots (88.17%).

ANOVA analysis indicated that both 6-BA and NAA had a highly significant effect on the callus induction rate of *P. japonicus* (*p* < 0.01). Range analysis showed that NAA had a more crucial role than 6-BA in root callus induction.

When the concentration of NAA is less than 16.11 μmol·L^−1^, callus induction by the root was lower, and the callus was dry. As the concentration of 6-BA increased, callus growth gradually increased. At 6-BA concentrations above 13.32 μmol·L^−1^, the callus became relatively dense, and the ability to proliferate was relatively weak in the later proliferation culture (Appendix A).

### 2.2. Proliferation Culture of Root Callus of Panax japonicus and Screening of High-Yield Cell Lines

#### 2.2.1. Effects of NAA and 6-BA on Callus Proliferation of *Panax japonicus*

As shown in Table 1, the proliferation rate and the condition of the callus varied with the changes in the medium composition. The optimal formula was MS + 1.0 g·L^−1^ MS + 4.64 μmol·L^−1^ NAA + 13.32 μmol·L^−1^ 6-BA + 30.0 g·L^−1^ sucrose + 3.8 g·L^−1^ gellan gum (Scheme 4 in Table 1). This combination produced callus with a loose structure and easy proliferation.

The ratio of NAA and 6-BA in the medium influenced the proliferation of the callus (Figure 1). Using the formulation from Scheme 4 (in Table 1), the concentration of 6-BA (13.32 μmol·L^−1^) was held constant and the mass concentration ratios of NAA and 6-BA in the medium were adjusted to 1:1, 1:2, 1:3, 1:4, 1:5 and 1:10, i.e., the concentration of NAA (16.11 μmol·L^−1^, 8.06 μmol·L^−1^, 5.37 μmol·L^−1^, 4.03 μmol·L^−1^, 3.22 μmol·L^−1^, 1.61 μmol·L^−1^) was changed. After 21 days of culture, it was observed that the ratio of NAA to 6-BA significantly affected the callus proliferation rate (*p* < 0.05). As the ratio was increased, the proliferation rate of the five cell lines generally increased and then decreased. With the exception of cell line L-2, the proliferation rate of the five cell lines peaked at 1:3. Therefore, an NAA to 6-BA ratio of 1:3 is suitable for callus proliferation.

At a ratio of NAA: 6-BA of 1:3, varying concentrations of NAA and 6-BA in the medium were tested: low concentration (2.69 μmol·L^−1^ NAA + 6.66 μmol·L^−1^ 6-BA), medium concentration (5.37 μmol·L^−1^ NAA + 13.32 μmol·L^−1^ 6-BA) and high concentration (8.06 μmol·L^−1^ NAA + 19.95 μmol·L^−1^ 6-BA). After 21 days of culture, it was found that both lower and higher hormone concentrations significantly inhibited callus proliferation (*p* < 0.05). The highest callus proliferation ability among the five cell lines was achieved at a moderate concentration (5.37 μmol·L^−1^ NAA + 13.32 μmol·L^−1^ 6-BA) (Figure 2).

According to the normal growth curve (Figure 3a) and the semi-logarithmic growth curve (Figure 3b) of the callus, the fresh weight of the callus of the five cell lines consistently increased with longer culture times. By the 21st day of culture, the growth slowed and entered a stationary growth phase. Therefore, subculturing should be performed on the 21st day of culture.

#### 2.2.2. Screening of High-Yielding Cell Lines of *Panax japonicus*

Figure 4 shows that the proliferation coefficient of cell line L-1 was significantly higher than that of cell lines L-2 and L-4, making it the fastest-growing cell line. Triterpenoid glycoside content varied among the cell lines (Figure 4), and levels in cell lines L-1 and L-2 were significantly higher than those in the other cell lines. Based on the comprehensive analysis of the proliferation coefficient and their content level of the five cell lines, it was found that cell line L-1 was a high-yielding cell line with rapid proliferation (proliferation coefficient of 2.88) and relatively high “total saponin” content (123.61 μg/g dry weight).

#### 2.2.3. Effect of Curing Agents on the Proliferation of *Panax japonicus* Callus Tissues

Figure 5 shows that the callus is looser and larger when the curing agent is gellan gum.

### 2.3. Establishment of Suspension Culture System of Panax japonicus

#### 2.3.1. Effect of Carbon Source on Cell Suspension Culture

The carbon sources had significant effects on the growth parameters—cell density, SCV, FW and DW—in the suspension culture (*p* < 0.05) but no significant effects on cell viability. Among the carbon sources tested, sucrose resulted in the fasted growth (Figure 6), followed by maltose, with glucose leading to the slowest growth.

#### 2.3.2. Effect of Initial Inoculum Size on Suspension Culture

The initial inoculation amount significantly affected the five growth parameters—cell density, viability, SCV, FW and DW—in the suspension culture (*p* < 0.01). As the initial inoculation amount increased, the cell density, SCV in the suspension culture medium, FW and DW of the culture first increased and then decreased (Figure 7). Although cell density, SCV and DW reached the highest values at an initial inoculation amount of 1.0 g, the multiplication rate was lower than at 0.8 g, achieving 47.69%.

#### 2.3.3. Effect of Shaking Speed on Suspension Culture

The shaking speed had a significant effect on cell density, SCV, FW and DW in the suspension culture (*p* < 0.05). With increasing shaking speed, cell density, SCV, FW and DW first increased and then decreased (Figure 8). The highest value was reached at 110 r/min.

#### 2.3.4. Effect of Plant Growth Regulators on Suspension Culture of Callus Tissues

The levels of NAA and 6-BA had significant differences in the cell density, SCV, FW and DW of the suspension culture (*p* < 0.05) (Figure 9). When the cell cultures were cultured in a medium containing higher or lower concentrations of hormones (NAA and 6-BA), the growth of the cell cultures was relatively slow, and the growth parameters were lower than at moderate concentrations.

#### 2.3.5. Effect of Culture Days on Suspension Culture

With increasing culture duration, cell density, SCV, FW and DW initially increased and then decreased (Figure 10). On day 21, each parameter reached its maximum value (cell density = 26.07 × 10^7^/mL, SCV = 4.9 mL, FW = 2.0546 g, DW = 0.1627 g), and cell viability began to decrease. With the extension of the culture time, the content of “total saponins” in the culture continuously increased, peaking on the 21st day (243.06 μg/g dry weight), after which it began to decrease.

The growth of the cells grown in suspension culture followed an S-shaped curve (Figure 11). Over the first 28 days of culture, cell growth could be divided into three distinct phases: 0 days to 7 days was the growth lag period, then the logarithmic growth phase followed, which lasted for 14 days and ended on the 21st day.

### 2.4. Effects of Suspension Culture Conditions on the Content of Total Saponins in Cells of Panax japonicus

#### 2.4.1. The Effect of Carbon Source on the Content of Triterpenoid Glycosides in Cells

The carbon source significantly influenced the triterpenoid glycoside content in the suspension culture (*p* < 0.05). When glucose was used as the carbon source (Figure 12), the “total saponins” content in the suspension culture was the highest (125.00 μg/g dry weight). Sucrose resulted in the second-highest ginsenoside content, while maltose yielded the lowest.

#### 2.4.2. Effect of Initial Inoculation on Total Saponins Content

The initial inoculation amount also significantly affected the total saponins content in the suspension culture (*p* < 0.05), and the content of total saponins increased, after which it decreased with the increase in the initial inoculation amount (Figure 13). At an initial inoculation amount of 0.8 g, the total saponins content in the culture was the highest (209.72 μg/g dry weight).

#### 2.4.3. Effect of Shaking Speed on the Content of Triterpenoid Glycosides

The shaking speed significantly influenced the triterpenoid glycoside content in the suspension culture (*p* < 0.05). As the rotation speed increased (Figure 14), the change in ginsenoside content corresponded to the growth trend of the suspension culture (first increased and then decreased), and the “total saponins” content achieved the maximum value (209.72 μg/g dry weight) at 110 r/min.

#### 2.4.4. Effects of Plant Growth Regulators on the Content of Triterpenoid Glycosides

The concentration of NAA and 6-BA significantly influenced the triterpenoid glycoside content in the suspension culture (*p* < 0.05). Both higher and lower concentrations of NAA and 6-BA inhibited ginsenoside accumulation in suspension cultures (Figure 15). The “total saponins” content (209.72 μg/g dry weight) of the suspension culture with a moderate concentration was significantly higher than the “total saponins” content of the cells cultured with a higher or lower concentration.

#### 2.4.5. The Effect of Culture Time on the Content of Triterpenoid Glycosides

With the extension of the culture time, triterpenoid glycoside content in the suspension culture showed a general upward trend (Figure 16). During the cell growth retardation phase, the ginsenoside content in the cells increased rapidly. In the first 7 days of the logarithmic growth phase (7 days to 14 days of culture), “total saponins” increased rapidly and then slowly increased until the level of “total saponins” rapidly decreased in the stationary growth phase.

## 3. Discussion

### 3.1. Callus Induction of Aseptic Roots of Panax japonicus

The key to a successful plant tissue culture is obtaining sterile explants [17]. The quality of the explants has significantly impacted the quality of subsequent experiments [18]. Contamination is the biggest problem in the process of plant tissue culture, often resulting from infected explants [19]. Luo et al. [12] disinfected the embryos, stems and leaves of *P. japonicus* to obtain sterile material. However, the disinfection of *P. japonicus* roots has not yet been investigated. Different disinfection methods were used in this work to treat *P. japonicus* roots. It was found that the lowest contamination rate was achieved by disinfecting the roots in 75% alcohol for 60 s, followed by a 10% NaClO treatment for 15 min.

Plant growth regulators are crucial for callus induction [20], with appropriate ratios and concentrations of growth factors and cytokinin significantly impacting the induction of healing tissue [21]. In this work, we changed the concentrations of NAA and 6-BA in the healing induction medium and found that the highest rate of healing induction was achieved when the concentration was 3 mg·L^−1^ for both NAA and 6-BA. The healing induction rate was higher than reported in previous studies [9,11,12]. Our findings indicate that both NAA and 6-BA significantly affect the healing induction of *P. japonicus*, with NAA playing a dominant role. Notably, no healing callus growth was observed in *P. japonicus* with no exogenous hormone added, consistent with the results by Zhu et al. [9]

### 3.2. Callus Culture and High-Yield Cell Line Screening of Panax japonicus

While the callus is easier to induce and maintain, loose healing tissue requires continuous subculturing to maintain it [22]. When the healing tissue was initially induced, the medium for healing proliferation should be adjusted according to the type of healing tissue, allowing loose healing tissue maintenance through subculturing [23]. In this study, by continuously adjusting the ratio of NAA and 6-BA in the medium for *P. japonicus* healing proliferation, loose healing tissue of *P. japonicus* was finally obtained. We also found that increasing the concentration of 6-BA made the callus of *P. japonicus* looser, which was consistent with the results of He et al. [24]. Rapid propagation of plant callus requires appropriate combinations and concentrations of plant growth regulators [25,26], as well as suitable culture durations to maintain and grow loose callus, such as of *P. ginseng*. Three to four weeks for subculturing the callus is sufficient [27]. In this work, by changing the ratio and concentration of NAA and 6-BA in the proliferation medium of *P. japonicus* and analyzing the growth curves of the healing tissues, we found that the fastest proliferation of the healing tissues was achieved with 5.37 μmol·L^−1^ NAA and 13.32 μmol·L^−1^ 6-BA, with an optimal culture duration of 3 weeks (21 d).

The most important characteristic of *P. japonicus* as a member of the *P. ginseng* family is its richness in triterpenoid glycosides. While its total saponins synthesis and metabolism have been studied [28,29], there are almost no studies on total saponins production from *P. japonicus* callus culture. In this study, we observed that callus growth and ginsenoside content varied among different cell lines, likely due to the growth age and genotype of *P. japonicus*. For the efficient proliferation of *P. japonicus* and the effective accumulation of biologically active ginsenosides, it is essential to screen cell lines for rapid proliferation and high “total saponins” content. In this study, we identified cell line L-1, which shows promise as a valuable test material for subsequent studies. Similar research has been conducted on cell lines for high production of anthocyanins and proanthocyanidins in *Vitis davidii* Foëx [30].

### 3.3. Panax japonicus Cell Suspension Culture System

Suspension cultures are typically initiated by suspending the callus in a liquid medium. There are many factors that affect suspension cultures, such as initial inoculation, and shaking speed, all of which influence the growth of suspension cultures and the accumulation of biologically active ginsenosides. Previous reports have highlighted the importance of initial inoculation density for the successful establishment of the suspension culture [31]. In the suspension culture of Western ginseng (*Panax quinquefolius* L.) [32], the fastest growth of callus was observed at an initial inoculation density of 50 g·L^−1^, while in the study of *P. japonicus*, it was found that the fastest proliferation of healing and the highest “total saponins” content were observed at an initial inoculation density of 40 g·L^−1^. In addition, the determination of the cell growth curve is crucial for optimal subculture timing and evaluating the performance at different growth stages [33]. In this work, we observed that the *P. japonicus* cells in suspension culture followed an S-shaped growth curve, reaching peak growth and total saponins content at 21 days, which was consistent with the results of Glagoleva [15]. During the suspension culture of *P. japonicus* cells, the cell mass increased, resulting in a lower cell density at the end of the culture compared to the beginning. Future work will focus on the suspension culture of single isolated cells or small cell clusters to enhance growth and total saponins production.

The rotation speed of the shaker is a crucial factor influencing suspension culture. A higher rotation speed increases mechanical damage to the cells, potentially leading to cell death; conversely, a lower rotation speed hinders the dispersion of cell clusters, preventing adequate nutrient absorption and slowing the growth rate [34]. In this study, different rotation speeds did not significantly affect cell viability but had a substantial impact on culture growth and total saponins accumulation. The maximum biomass and total saponins accumulation were observed at a rotation speed of 110 rpm.

Furthermore, both high and low concentrations of hormone additions inhibited the growth of *P. japonicus* suspension cultures, which contrasts with previous studies that suggested liquid cultures require lower hormone levels than solid cultures [35]. This discrepancy may be due to the excessive difference in the establishment of the hormone gradient and needs to be further investigated.

The growth of almost all plant cell cultures depends on the carbon source in the medium, in addition to phytohormones. In this study, the effects of three carbon sources (sucrose, maltose and glucose) on callus growth and triterpenoid glycoside formation were compared. It was found that sucrose was the most favorable for *P. japonicus* callus growth, while growth was retarded by glucose and, to a lesser extent, by maltose. This situation is probably due to the specificity of the sugar uptake in ginseng cells in vitro. In contrast to many plant cell cultures, which absorb whole sucrose molecules by the use of sucrose transporters, in ginseng cells, enzymes for sucrose and maltose degradation are present in the cell wall and/or in culture media [36]. The hydrolysis of sucrose occurs extracellularly, and the resulting monosaccharides are absorbed by the cells with different efficiencies. In some cases, cells use all glucose available until glucose exhaustion before fructose is utilized [37]; in other cases, fructose is taken up more efficiently than glucose [A. Nosov et al., unpublished data].

In this study, we found that the effect of suspension culture conditions on the content of triterpenoid glycosides in *P. japonicus* suspension culture cells was consistent with their growth intensity. Changes in the content of “total saponins” over time coincided with changes in growth. This relationship may be specific to *P. japonicus*, as a similar phenomenon was observed in the study by Glagoleva et al. [15].

Under normal conditions, the glucose is sterilized separately by filtering. In our case, we did not use glucose but rather the product of degradation. In our studies, it was found that the degradation product from glucose in the medium reduced the growth intensity of the culture but increased the level of triterpenoid glycoside content. We know that the glucose is degraded into 5-hydroxymethylfurfural (5-HMF), acetic acid and formic acid under acidic conditions. The former is an intermediate product that can be further degraded, while the latter two, acetic acid and formic acid, are the final products of degradation. When the sterilization temperature of glucose was too high (121 °C), 5-HMF was produced. 5-HMF has certain side effects on growth. The use of maltose in the medium, the hydrolysis of which leads to the formation of two glucose molecules, leads to both a decrease in the growth of the culture and a significant decrease in the level of accumulation of ginsenosides. The noted patterns may be a consequence of the above-mentioned specificity of carbohydrate uptake by ginseng cells in vitro, particularly with a possible slower hydrolysis of maltose compared to sucrose [36]. Investigation of these patterns requires further studies.

## 4. Materials and Methods

### 4.1. Materials

Wild *P. japonicus* plants aged 2–30 years were collected in July 2022, March, May and June 2023 from dapping turnips of *P. japonicus* in Nayong County, Bijie City, Guizhou Province, and their roots were used as test materials.

### 4.2. Methods

#### 4.2.1. Callus Induction of *Panax japonicus*

##### Methods of Root Disinfection

After the bottom of the root epidermis was brushed and cleaned, it was placed on an ultra-clean bench. After disinfection with 75% alcohol for 60 s (changed every 30 s), it was disinfected with different concentrations (3%, 5%, 10%, *v*/*v*) of sodium hypochlorite (NaClO) for different times (8 min, 10 min, 12 min, 15 min, 18 min, 20 min), with a total of 12 treatments. The treated roots were rinsed 5–6 times with sterile water and cut into 0.5 cm thick slices, which were inoculated into a callus induction medium (MS + 16.11 μmol·L^−1^ NAA + 0.44 μmol·L^−1^ 6-BA + 30.0 g·L^−1^ sucrose+ 7.5 g·L^−1^ agar) for cultivation. There were three replicates per treatment. After adjusting pH of medium to 5.8, we carried out high-temperature and high-pressure sterilization (121 °C, 20 min). After inoculating, the cultural dishes with explant were cultured in the dark at 23 ± 2 °C. After 7 days, the infection rate, mortality and survival rate of the roots were counted.

##### Callus Induction Culture

The sterile roots were inoculated into the induction medium (MS) supplemented with different concentrations of NAA (0 μmol·L^−1^, 8.06 μmol·L^−1^, 16.11 μmol·L^−1^ and 32.22 μmol·L^−1^) and 6-BA (0 μmol·L^−1^, 6.66 μmol·L^−1^, 13.32 μmol·L^−1^ and 26.64 μmol·L^−1^). The L16 (24) orthogonal table design experiment was used, a total of 16 treatments, each culture dish with 5 explants, each treatment with 10 culture dishes, each group with 3 repeated tests. The sterilization method for the medium and cultivation conditions are the same as in the section Methods of Root Disinfection. In the culture, the induction of callus was observed every 3 days, the condition of callus was observed after 30 days of culture, the induction rate of callus was counted and the range analysis was performed to find out the most suitable medium for callus induction.

##### Computing Formula

The contamination rate, survival rate, mortality rate and callus induction rate of the explants were expressed as a percentage. The percentage was sinusoidally inverted before statistical analysis, and then a one-way analysis of variance (IBM SPSS Statistics 25) was performed.
(1)$$Death\;rate\;(\%)=\frac{Number\;of\;deathe\;explants}{Number\;of\;inoculated\;explants}\times 100$$
(2)$$Survival\;rate\;(\%)=\frac{Number\;of\;survival\;explants}{Number\;of\;inoculated\;explants}\times 100$$
(3)$$Callus\;induction\;rate\;(\%)=\frac{Number\;of\;explants\;for\;induction\;of\;calls}{Number\;of\;inoculated\;explants}\times 100$$

#### 4.2.2. Proliferation Culture of Root Callus of *Panax japonicus* and Screening of High-Yield Cell Lines

##### Proliferation Culture of Loose Callus Tissue

The cell line L-1 was used as experimental material and the callus was transferred to the proliferation medium (MS + 16.11 μmol·L^−1^ NAA + 0.44 μmol·L^−1^ 6-BA + 30.0 g·L^−1^ sucrose + 3.8 g·L^−1^ gellan gum [11]). Subculture was performed every 21 days, and the subculture medium was the same as the proliferation medium. Observe the condition of the callus and adjust the concentration of exogenous hormones in the medium to make the callus looser.

Using L-1, L-2, L-3, L-4 and L-5 as the experimental material, the ratio of NAA and 6-BA in the optimized proliferation medium (1:1, 1:2, 1:3, 1:4, 1:5, 1:10) was changed and the fresh weight of the callus of the five cell lines on the 21st day of the experiment was measured. The effects of the ratio of NAA and 6-BA on callus growth were analyzed and compared, the appropriate ratio of NAA and 6-BA for callus proliferation of P. japonicus was determined and the medium for callus proliferation of *Panax japonicus* was optimized. Each treatment had three replicates.

##### Culture Cycle of Callus

The loose callus of cell lines L-1, L-2, L-3, L-4 and L-5 was selected for proliferation culture. The fresh weight of the callus of five cell lines was weighed after 0 day, 3 days, 6 days, 9 days, 12 days, 15 days, 18 days, 21 days, 24 days and 27 days of culture, respectively. The growth curve of the callus was plotted, and the subculture time of the callus was observed and determined.

##### Screening of High-Yield Cell Lines

Using the cell lines L-1, L-2, L-3, L-4 and L-5 as experimental material, 1.0 g of callus was weighed and placed on the surface of the selected proliferation medium. At the end of a culture cycle, the fresh weight of the callus was weighed and then dried at 60 °C to constant weight, and its dry weight was weighed. At the same time, the content of “total saponins” in the callus was determined, and the cell lines with rapid growth and high content of “total saponins” were selected.

The “total saponins” determination method: The “total saponins” determination kit (Suzhou Grease Biotechnology Co., Ltd., Suzhou, China) was used for the determination. After the callus was dried, crushed and sieved, 0.05 g was weighed and placed in an EP tube; then, 1 mL of extract was added, ultrasonically extracted for 1 h and then centrifuged at 25 °C for 10 min at 8000× *g*. A total of 0.5 mL of the supernatant was taken for assay. Before determination, the supernatant was evaporated to dryness at 70 °C, 0.2 mL of reagent 1 and 0.8 mL of reagent 2 (perchloric acid) were added, and the water bath was set at 55 °C for 20 min. A total of 200 μL was added to a 1 mL glass cuvette, and then 1000 μL of reagent Tris (acetic acid) was added. After complete mixing, absorbance A1 was measured at 589 nm. The blank tube was 0.5 mL of extract, the treatment method was the same as that of the determination tube, and the absorbance was A2. Calculate ΔA = A1 − A2. The visible spectrophotometer is preheated for more than 30 min, the wavelength is set to 589 nm and the distilled water is set to zero.

##### Screening of Curing Agent

The loose callus was selected for proliferation culture, the type of curing agent (gellan gum, agar) in the proliferation medium was changed and the growth status of the callus was compared at the end of a culture cycle to find out the type of curing agent suitable for callus proliferation culture of *P. japonicus*.

##### Computing Formula

The calculation of callus proliferation and total saponins content of *P. japonicus* is shown in the following formula:(4)$$Callus\;induction\;rate\;(\%)=\frac{Number\;of\;explants\;for\;induction\;of\;calls}{Number\;of\;inoculated\;explants}\times 100$$
(5)$$Callus\;proliferation\;coefficient=\frac{\;FW\;of\;callus\;after\;proliferation\;(g)}{FW\;of\;callus\;before\;proliferation\;(g)}$$
(6)$$Total\;Saponin\;Content\left(\frac{\mu g}{gDW}\right)=\frac{\frac{\Delta A+0.012}{0.0072}\times V1}{\frac{V1}{V2}\times W}=\frac{138.89\left(\Delta A+0.012\right)}{W}$$

Note: *V1:* add sample volume, 0.5 mL; *V2*: add the volume of the extract, 1 mL; *W*: sample dry weight, g.

### 4.3. Establishment of Cell Suspension Culture System of Panax japonicus

#### 4.3.1. Screening of Suspension Culture Conditions

A total of 1.0 g tissue fresh weight (FW) was taken from the periphery of the loose callus and suspended in 20 mL liquid proliferation medium (MS + 5.37 μmol·L^−1^ NAA + 13.32 μmol·L^−1^ 6-BA + 30.0 g·L^−1^ sucrose). After shaking well, it was poured into a 100 mL Erlenmeyer flask. The Erlenmeyer flask was sealed with aluminum foil and sealing foil and placed on a horizontal orbital oscillator (rotation radius of 30 mm) for culture at 110 r/min in the dark at 25 ± 1 °C.

#### 4.3.2. Carbon Source Screening

After 7 days of culture, the types of carbon sources (sucrose, maltose, glucose) were changed in the liquid medium and cultivated at 110 r/min in the dark at 25 ± 1 °C.

#### 4.3.3. Screening of Initial Inoculation Amount

When cultured for 7 days, the initial inoculation amount was changed to 0.2 g/20 mL, 0.4 g/20 mL, 0.6 g/20 mL, 0.8 g/20 mL, 1.0 g/20 mL, 1.2 g/20 mL, 1.4 g/20 mL.

#### 4.3.4. Cell Rotation Speed Screening

At 7 days of culture, the shaking speed (100 r/min, 110 r/min, 120 r/min) was changed.

#### 4.3.5. Hormone Concentration Screening

Based on the existing solid culture hormone combination of 5.37 μM NAA + 13.32 μM 6-BA, the hormone concentration was changed, i.e., lower concentration (2.32 μmol·L^−1^ NAA + 6.66 μmol·L^−1^ 6-BA), medium concentration (5.37 μmol·L^−1^ NAA + 13.32 μmol·L^−1^ 6-BA) and higher concentration (5.37 μmol·L^−1^ NAA + 19.98 μmol·L^−1^ 6-BA).

The above (1)–(4) experiments were performed on the 21st day of culture, and the cell density, viability, suspension sedimentation volume (SCV), fresh weight (FW) and dry weight (DW) of the cultured suspension cultures were measured. Each treatment was repeated three times.

#### 4.3.6. Screening of Incubation Time

After 7 days of culture, the culture was performed under the optimal culture conditions determined in the above-mentioned experiments. The cell density, viability, SCV, FW and DW of the cultured suspension cultures were determined at 0 day, 7 days, 14 days, 21 days and 28 days of culture with three replicates for each treatment.

#### 4.3.7. Methods for Determination of Cell Growth Index

##### Determination of Cell Viability and Density

The cell counting plate method was used to observe the cells and to calculate cell viability and density. A 1:1 mixture of the cell suspension is to be analyzed, the Trypan blue staining solution is added drop by drop to a cell counting plate and the total number of cells in the four main compartments is counted. Trypan blue stains dead cells blue, while live cells are not stained to distinguish live cells from dead cells. The viability of the cells can then be calculated.
(7)$$Cell\;density\;(N/mL)=\frac{Total\;number\;of\;cells\;in\;four\;large\;squares\times 2}{4}\times 10,000$$
(8)$$Vitality\;(\%)=\frac{Number\;of\;living\;cells\;}{Total\;number\;of\;cells\;}\times 100$$

##### Determination of Settled Cell Volume

The culture medium to be tested was poured into a 20 mL graduated tube and allowed to stand for 30 min. After all cultures had settled, the volume of the culture was measured, i.e., the settled cell volume (SCV).

##### Determination of Biomass

The suspension culture solution was placed in a Buchner funnel, the suspension culture solution was filtered under a low-pressure pulse onto a pre-weighed filter paper disc and its fresh weight (FW) was weighed. The culture was then oven-dried at 60 °C for 24 h to determine the dry weight (DW).

#### 4.3.8. Data Analysis Methods

Three replicates of each treatment were used, and the results of the experiment were subjected to one-way analysis of variance (ANOVA) to analyze and compare the effects of different culture conditions on the suspension culture of *P. japonicus* healing tissues.

### 4.4. Detection of Total Saponins Content in Suspension-Cultured Cells

The healing tissue cultures under different culture conditions in Section 4.3. were taken and dried to constant weight and then their saponins were determined using the method of total saponin determination in the section Screening of High-Yield Cell Lines. A one-way analysis of variance (ANOVA) was also performed to compare the effects of different culture conditions on the accumulation of total saponins in *P. japonicus*.

## 5. Conclusions

This work established a callus induction and suspension culture system for the roots of *P. japonicus*, including the following steps: disinfection of the root in 75% alcohol for 60 s and then treatment in 10% NaClO for 15 min; callus induction of the roots in the medium MS + 16.11 μmol·L^−1^ NAA + 13.32 μmol·L^−1^ 6-BA + 30.0 g·L^−1^ sucrose + 7.8 g·L^−1^ agar; callus inoculation in MS +5.37 μmol·L^−1^ NAA + 13.32 μmol·L^−1^ 6-BA + 30.0 g·L^−1^ sucrose + 3.8 g·L^−1^ gellan gum, subcultured every 21 days; establishment of suspension culture by transferring the callus on growth medium (MS + 5.37 μmol·L^−1^ NAA + 13.32 μmol·L^−1^ 6-BA + 30.0 g·L^−1^ sucrose) and cultured in a shaker at 110 r/min for 21 days; replacing the growth medium with the production medium (MS + 5.37 μmol·L^−1^ NAA + 13.32 μmol·L^−1^ 6-BA + 30.0 g·L^−1^ glucose) for maximum accumulation of “total saponins”. The established culture system can promote the accumulation of total saponins in callus cells and provide a basis for large-scale cell culture and industrial production of medicinal “total saponins”.

## Figures and Tables

**Figure 1 plants-13-02480-f001:**
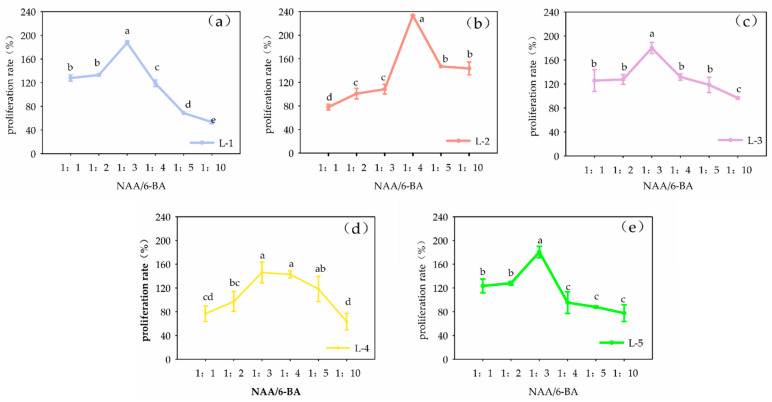
Effects of NAA/6-BA on callus proliferation of different cell lines of *Panax japonicus.* Note: Different lowercase letters indicate that different ratios of NAA and 6-BA have significant effects on callus proliferation (*p* < 0.05). Note: (**a**) is the effect of NAA/6-BA on the growth of L-1 cell line, (**b**) is the effect of NAA/6-BA on the growth of L-2 cell line, (**c**) is the effect of NAA/6-BA on the growth of L-3 cell line, (**d**) is shows the effect of NAA/6-BA on the growth of L-4 cell line, (**e**) is the effect of NAA/6-BA on the growth of L-5 cell line.

**Figure 2 plants-13-02480-f002:**
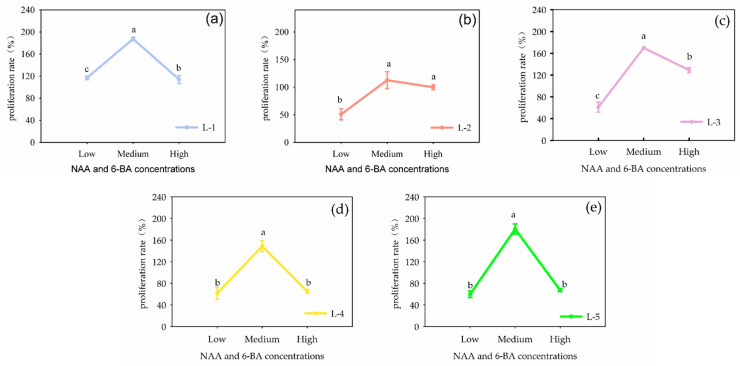
Effects of the concentration of NAA and 6-BA on *Panax japonicus* callus proliferation in different cells. Note: Different lowercase letters indicated that different concentrations of NAA and 6-BA had significant effects on callus proliferation (*p* < 0.05). Note: (**a**) is the effects of the concentration of NAA and 6-BA on L-1, (**b**) is the effects of the concentration of NAA and 6-BA on L-2, (**c**) is the effects of the concentration of NAA and 6-BA on L-3, (**d**) is the effects of the concentration of NAA and 6-BA on L-4, (**e**) is the effects of the concentration of NAA and 6-BA on L-5.

**Figure 3 plants-13-02480-f003:**
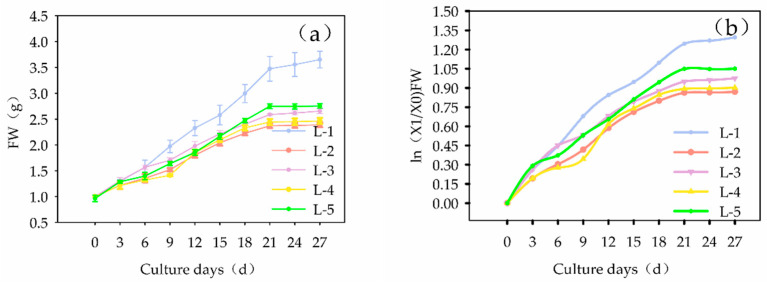
Normal and semi-logarithmic growth curves of callus of different cell lines of *Panax japonicus*. Note: (**a**) Normal growth curve of callus of different cell lines, (**b**) semi-logarithmic growth curve of callus of different cell lines.

**Figure 4 plants-13-02480-f004:**
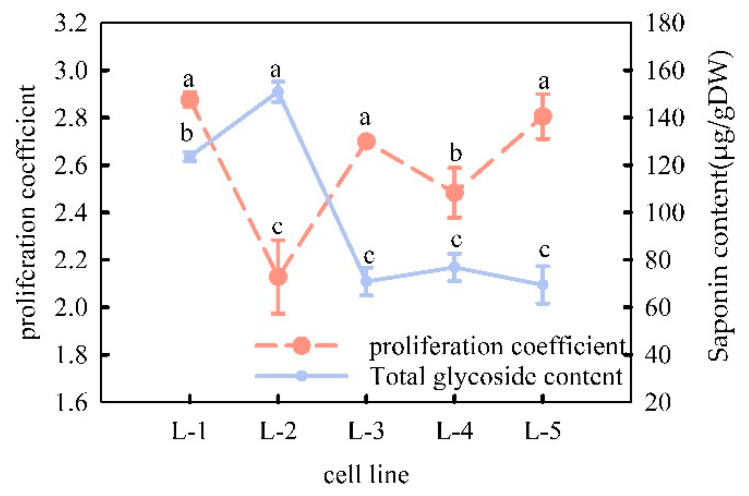
Comparison of proliferation coefficient and “total saponins” content of different cell lines of *Panax japonicus*. Note: different lowercase letters indicate significant differences in proliferation coefficients as well as saponin content between cell lines (*p* < 0.05).

**Figure 5 plants-13-02480-f005:**
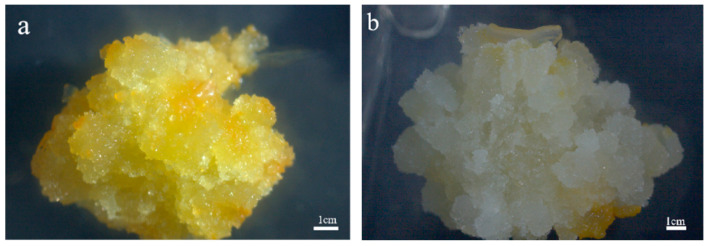
Effects of different curing agents on the growth of callus of *P. japonicus* L-1 cell line (cultured for 21 days). Note: (**a**) is the callus when the curing agent is agar, (**b**) is the callus when the curing agent is gellan gum.

**Figure 6 plants-13-02480-f006:**
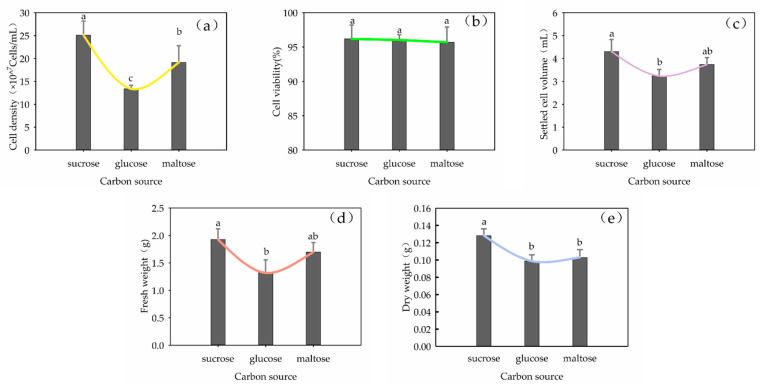
Effect of carbon source on cell growth of *Panax japonicus* suspension culture. Note: (**a**) is the effect of carbon source on cell density; (**b**) is the effect of carbon source on cell viability; (**c**) is the effect of carbon source on suspension sedimentation volume; (**d**) is the effect of carbon source on fresh weight; (**e**) is the effect of carbon source on dry weight. Different lowercase letters indicate that different carbon sources had significant effects on the growth of *P. japonicus* cells. (*p* < 0.05).

**Figure 7 plants-13-02480-f007:**
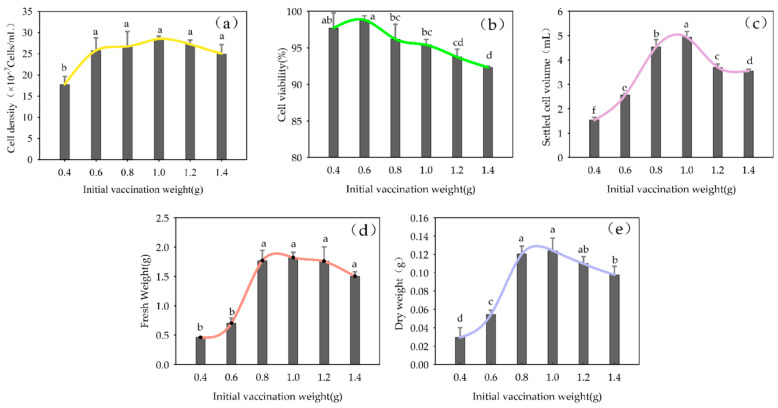
The effect of initial inoculation amount on cell growth in suspension culture of *Panax japonicus* Note: (**a**) is the effect of initial weight on cell density; (**b**) is the effect of initial weight on cell viability; (**c**) is the effect of initial contact weight on suspension colonization volume; (**d**) is the effect of initial weight on fresh weight; (**e**) is the effect of initial weight on dry weight. Different lowercase letters indicated that different amounts of initial inoculation had significant effects on the growth of *P. japonicus* cells (*p* < 0.05).

**Figure 8 plants-13-02480-f008:**
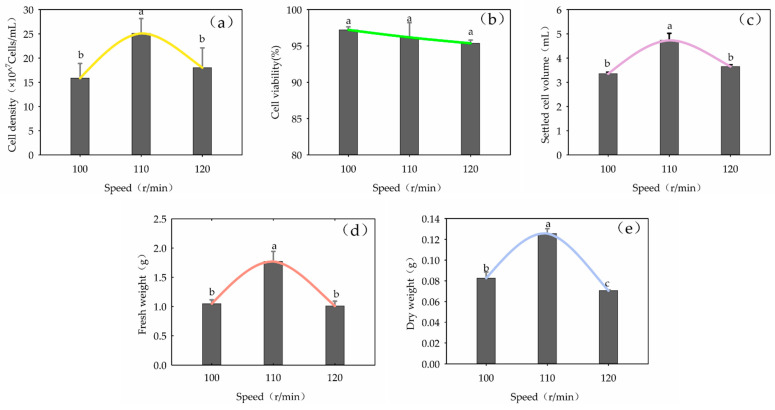
Influence of shaking speed on cell growth in the suspension culture of *Panax japonicus*. Note: (**a**) is the effect of shaking speed on cell density; (**b**) is the effect of shaking speed on cell viability; (**c**) is the effect of shaker rotation speed on suspension colonization volume; (**d**) is the effect of shaking speed on fresh weight; (**e**) is the effect of shaking speed on dry weight. Different lowercase letters indicated that different shaking speeds had a significant effect on the cell growth of *P. japonicus* (*p* < 0.05).

**Figure 9 plants-13-02480-f009:**
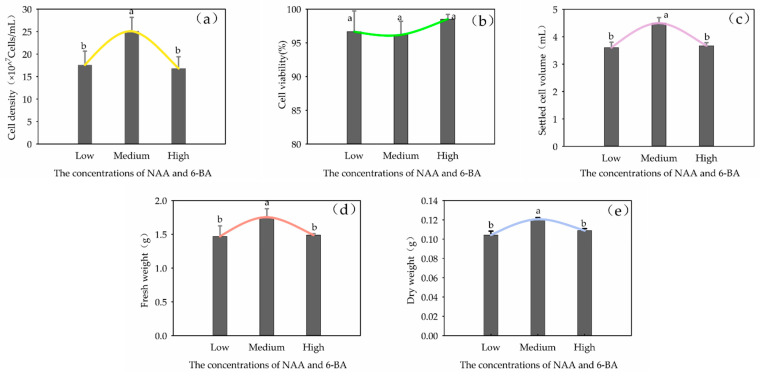
Effects of concentrations of NAA and 6-BA on cell growth in suspension culture of *Panax japonicus*. Note: (**a**) is the effect of NAA and 6-BA concentration on cell density; (**b**) is the effect of NAA and 6-BA concentration on cell viability; (**c**) is the effect of NAA and 6-BA concentration on suspension sedimentation volume; (**d**) is the effect of NAA and 6-BA concentration on fresh weight; (**e**) is the effect of NAA and 6-BA concentration on dry weight; different lowercase letters indicate that NAA and 6-BA concentration had a significant effect on cell growth of *P. japonicus* (*p* < 0.05).

**Figure 10 plants-13-02480-f010:**
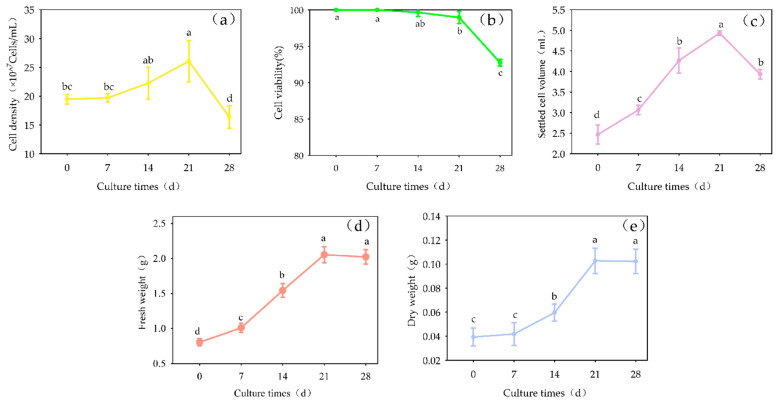
Growth curve of cell suspension culture of *Panax japonicus*. Note: (**a**) is the change curve of cell density; (**b**) is the cell density curve; (**c**) is the volume change curve of suspension colonization; (**d**) is the change curve of fresh weight; (**e**) is the change curve of dry weight. Different lowercase letters indicate that there is a significant difference in the growth of *Panax japonicus* cells on different culture days (*p* < 0.05).

**Figure 11 plants-13-02480-f011:**
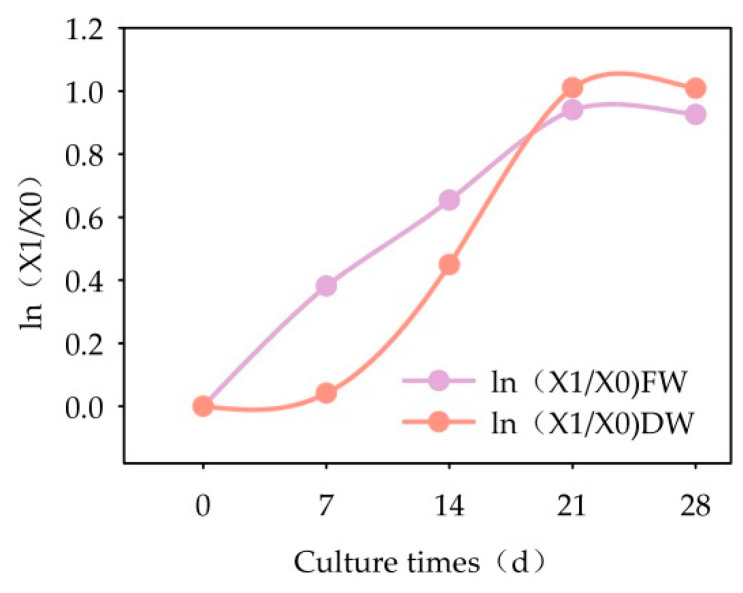
Semi-logarithmic growth curve of suspension-cultured cells of *Panax japonicus*.

**Figure 12 plants-13-02480-f012:**
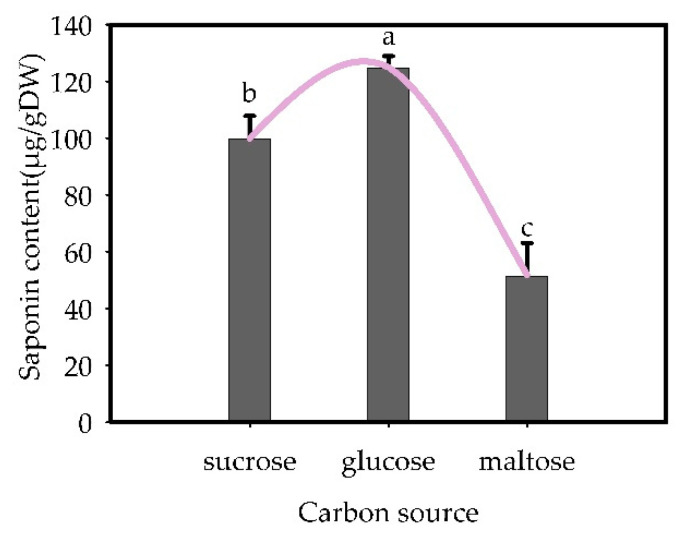
Effects of carbon sources on the content of “total saponins” in suspension-cultured cells of *Panax japonicus*. Note: Different lowercase letters indicate that different carbon sources have significant effects on “total saponins” in *Panax japonicus* (*p* < 0.05).

**Figure 13 plants-13-02480-f013:**
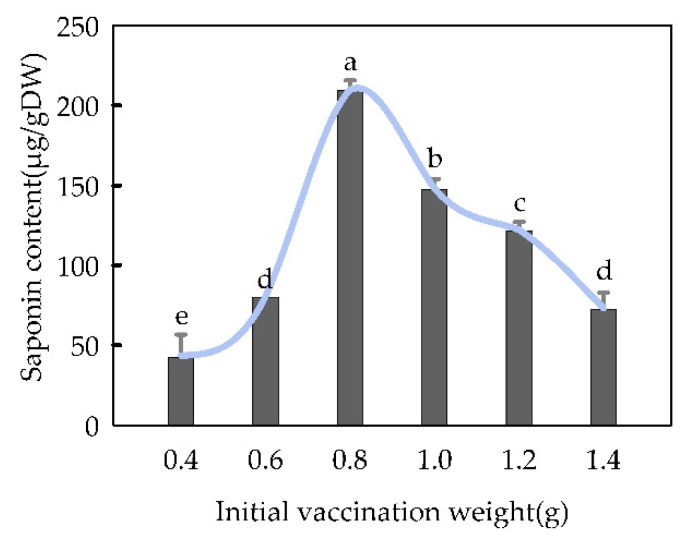
Effect of initial grafting weight on “total saponins” content in *Panax japonicus* cells cultured in suspension. Note: Different lowercase letters indicate that the different amount of initial inoculation has a significant effect on “total saponins” in *Panax japonicus* (*p* < 0.05).

**Figure 14 plants-13-02480-f014:**
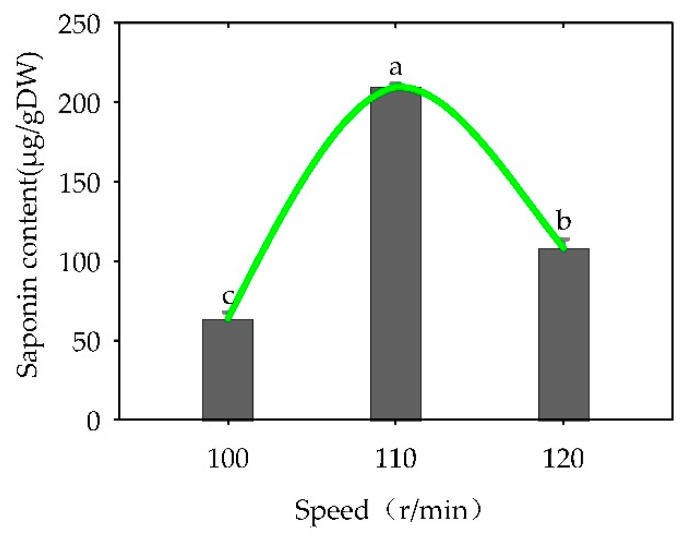
Effect of shaking speed on the content of “total saponins” in *Panax japonicus* cells cultivated in suspension. Note: Different lowercase letters indicate that different shaking speeds have a significant effect on “total saponins” in *Panax japonicus* (*p* < 0.05).

**Figure 15 plants-13-02480-f015:**
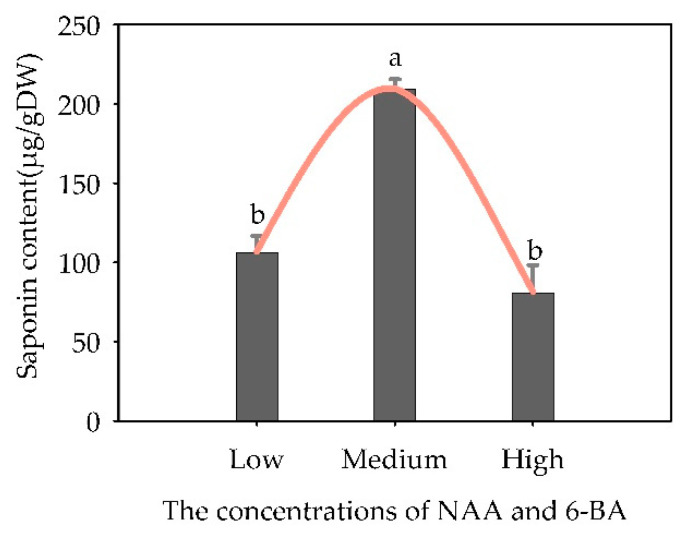
Effects of NAA and 6-BA concentrations on “total saponins” content in suspended cells of *Panax japonicus*. Note: Different lowercase letters indicate significant effects of different concentrations of NAA and 6-BA on “total saponins” of suspended cells of *Panax japonicus* (*p* < 0.05). Note: the lower concentration (2.32 μmol·L^−1^ NAA + 6.66 μmol·L^−1^ 6-BA), medium concentration (5.37 μM NAA + 13.32 μM 6-BA) and higher concentration (5.37 μM NAA + 19.98 μM 6-BA); see Section 4.3.5.

**Figure 16 plants-13-02480-f016:**
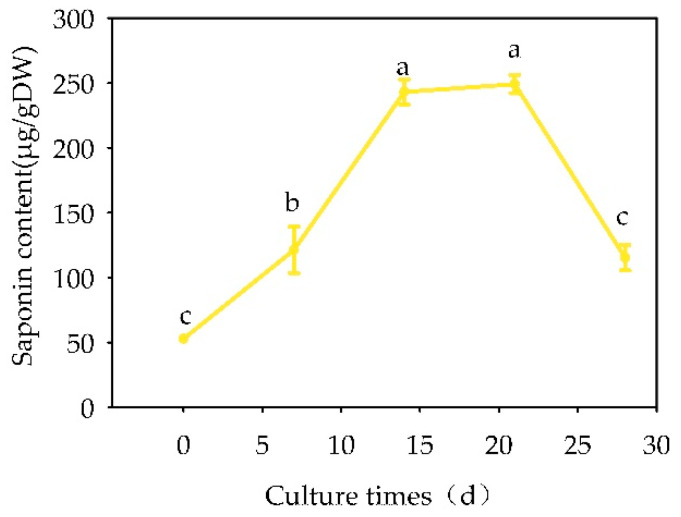
Influence of cultivation days on the “total saponins” content in the callus tissue of *Panax japonicus* suspension cells. Note: Different lowercase letters indicate significant differences in the “total saponins” content of the suspended cells of *Panax japonicus* during the different cultivation days (*p* < 0.05).

**Table 1 plants-13-02480-t001:** Proliferation and condition of *Panax japonicus* callus on different proliferation.

Scheme	Medium	Proliferation Rate/%	State of the Callus
1	MS + 16.11 μmol·L^−1^ NAA + 0.44 μmol·L^−1^ 6-BA + 30.0 g·L^−1^ sucrose + 3.8 g·L^−1^ gellan gum	43.33	Yellow and dense structure and not agglomerated (Appendix A)
2	MS + 16.11 μmol·L^−1^ NAA +4.44 μmol·L^−1^ 6-BA + 30.0 g·L^−1^ sucrose + 3.8 L^−1^ gellan gum	126.55	White and transparent; the structure is relatively dense (Appendix A)
3	MS + 13.32 μmol·L^−1^ 6-BA + 30.0 g·L^−1^ sucrose + 3.8 g·L^−1^ gellan gum	57.53	White and transparent; relatively loose structure (Appendix A)
4	MS + 4.64 μmol·L^−1^ NAA + 13.32 μmol·L^−1^ 6-BA + 30.0 g·L^−1^ sucrose + 3.8 g·L^−1^ gellan gum	148.49	White and transparent; loose structure (Appendix A)

## Data Availability

The data presented in this study are available upon reasonable request from the corresponding author.

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
