# Peer review of "Induction and Suspension Culture of Panax japonicus Callus Tissue for the Production of Secondary Metabolic Active Substances"

_plants, 2024, doi:10.3390/plants13172480_

Round 1

Reviewer 1 Report

Comments and Suggestions for Authors

I have a few remarks to the Authors:

-       The methods are not adequately described.

1. The methodological part does not contain information: what statistical methods were used, how many repetitions of the experiments were carried out.

2. Many results were given in '%' , so how were the results interpolated?

- The results are not clearly presented.

3. For better transparency of the obtained relationships, a two-factor analysis should be performed - for example, for the results presented in the Figure 1. Two-factors analysis could indicate statistically significant differences in the responses of the five cells lines tested to the growth regulator systems used. It would be shown whether there is a significant 'Line x growth regulator combination' interaction. A similar situation applies to the results presented in Figures 2, 6, 7, 8, 9, 10.

4. What type of analysis of variance was performed for the results presented in 2.4 subsection? The authors should standardize the unit in which they report the concentrations of plant growth regulators μM - mg/L – for example Table 1 ‘MS+16.11 μM NAA+0.1 mg/ L 6-BA+30.0 g/Lsucrose+3.8 g/L gellan gum’.

5. Instead of mg/L should be used  mg dm−3, g dm−3.

6. The summary of the work is written inconsistently.

7. The editorial note relates to the fact that the Latin name of the species should be written in italics in the work.

Author Response

Response to Reviewer 1 Comments

Dear Reviewer,

We are very grateful to you for taking the time to read and modify our article again. We find that your comments play a very important role in improving the quality of our papers. We have carefully revised the paper in light of your comments, and please find our response to the comments made below. We marked the modified part of the manuscript in red.

Thank you for considering our revised manuscript!

Point 1:The methodological part does not contain information: what statistical methods were used, how many repetitions of the experiments were carried out.

Response 1: Thank you very much for your suggestion. We read the manuscript carefully and accepted your suggestions. We added to the analytical methods as well as the number of replicates.For details please see the Materials and Methods

Point 2: Many results were given in '%' , so how were the results interpolated?

- The results are not clearly presented.

Response 2: Thank you very much for your suggestion. We read the manuscript carefully and accepted your suggestions. For details please see the 4.2.1.3

Point 3: For better transparency of the obtained relationships, a two-factor analysis should be performed - for example, for the results presented in the Figure 1. Two-factors analysis could indicate statistically significant differences in the responses of the five cells lines tested to the growth regulator systems used. It would be shown whether there is a significant 'Line x growth regulator combination' interaction. A similar situation applies to the results presented in Figures 2, 6, 7, 8, 9, 10.

Response 3:Thank you very much for your suggestion.We did not consider the relationship between cell lines in analyzing the results of Fig. 1, but only screened for plant growth regulators, and the existence of significant differences between cell lines is reflected in the screening of high-yielding cells (i.e., Fig. 4). For Fig.6, 7, 8, 9, 10 in which there is only one cell line, there is only one variable and therefore a two-factor analysis could not be performed.

Point 4: What type of analysis of variance was performed for the results presented in 2.4 subsection? The authors should standardize the unit in which they report the concentrations of plant growth regulators μM - mg/L – for example Table 1 ‘MS+16.11 μM NAA+0.1 mg/ L 6-BA+30.0 g/Lsucrose+3.8 g/L gellan gum’.

Response 4: Thank you very much for your suggestion. We read the manuscript carefully and accepted your suggestions. For the results in subsection 2.4 we performed a one-way ANOVA and the research methods are described in lines 509-511. In addition to this we harmonized the plant growth regulator's by μmol L−1.

Point 5: Instead of mg/L should be used mg dm−3, g dm−3.

Response 5: Response 5: Thank you very much for your suggestion. We read the manuscript carefully and modified the mg/L to the mg L−1.

Point 6: The summary of the work is written inconsistently.

Response 6: Thank you very much for your suggestion. We read the manuscript carefully and accepted your suggestions.See the abstract section 40-42 lines for details.

Point 7:  The editorial note relates to the fact that the Latin name of the species should be written in italics in the work.

Response 7: Thank you very much for your suggestion. We read the manuscript carefully and accepted your suggestions. Latin names of species in the text have been changed to italics.

Reviewer 2 Report

Comments and Suggestions for Authors

Comments and Suggestions for Authors

In this study, an attempt was made to establish callus culture and suspension culture system from Panax japonicus, a valuable medicinal plant in China. The main medicinal properties of P. japonicus are anti-tumor, anti-diabetic, anti-aging and anti-inflammatory, as a result of its main medicinal ingredients – glycosides. The establishment of the cited cultures can promote the accumulation of glycosides in callus cells and serves as a basis for large-scale cell culture and industrial production of medicinal glycosides in light of the plant's limited resources and the long natural growth cycle and low seedling rate of traditional propagation methods with seeds and rhizomes

The manuscript follows the structure recommended by Plants. The applied research design and research methods are appropriate to achieve the purpose of the study. The results are well presented and supported with good quality tables and figures. Furthermore, the results were processed through quantitative and statistical analysis.

The following remarks and suggestions can be made:

Abstract

-          The sentence in line 27-30: “The conditions of suspension culture …” does not contain essential information and can be omitted, and the culture conditions listed in parentheses can be placed in the previous sentence, line 25: “… culture conditions (initial inoculation amount, carbon source, shaking speed, hormone concentration, culture time),…“

-          In line 30: “We found that the lowest (5) contamination rate…”, (5) should be placed after “rate”, and also units of measurement added – 5?

-          In line 32: “The highest (88.17%) induction rate…”, (88.17%) should be placed after “rate”

-          In line 33: to add “induced” after “callus”: “The callus induced was inoculated onto …”

-          (2) /line 33/, (3) /line 35/ and [Conclusion] /line 40/ to be omitted

Introduction

-          I suggest adding an introductory sentence in line 54 after "..risk of extinction [7]": “Plant tissue culture and cell suspension culture are two techniques that can provide alternative methods for rapid cultivation”

-          In line 57: “Zhu [9] et al.”, to be “Zhu et al. [9]” The same inaccuracy was comet further on, on line 307, and in other citation as: Luo [12] et al. /line 294/- to be Luo et al. [12], Glagoleva [14], et al./line 374/ - to be Glagoleva, et al. [14]

Materials and Methods

-          Line 377: “Wild P. japonicus aged 2-30 years…”, add “plants”: “Wild P. japonicus plants /aged 2-30 years/…”,

Conclusions

-          The first part of this section /lines 514-525/ needs to be edited as it sounds like a result, and my suggestion is to start with a sentence from line 526: This work established a callus induction and suspension culture system for the roots of P .japonicus, including the following steps: disinfection of the root in 75% alcohol for 60 seconds and then treatment in 10% NaClO for 15 minutes; callus  induction of the roots in the medium MS + 16.11 μM  NAA + 13.32 μM 6-BA + 30.0 g / L sucrose + 7.8 g / L agar;  callus inoculation in MS +  5.37 μM NAA + 13.32 μM 6-BA + 30.0 g / L sucrose + 3.8 g / L gellan gum, subcultured every 21 days; establishment of suspension culture by transferring the callus on growth medium (MS + 5.37 μM NAA + 13.32 μM 6-BA + 30.0 g / L sucrose) and cultured in a shaker at 110 r / min for 21 days; replacing the growth medium with the production medium (MS 524 + 5.37 μM NAA + 13.32 μM 6-BA + 30.0 g / L glucose) for maximum accumulation of total glycosides. The established culture system can promote the accumulation of glycosides in callus cells and provide a basis for large-scale cell culture and industrial production of medicinal glycosides.

 In conclusion, this manuscript is recommended for publication in “Plants”.

Author Response

Response to Reviewer 2 Comments

Dear Reviewer,

We are very grateful to you for taking the time to read and modify our article again. We find that your comments play a very important role in improving the quality of our papers. We have carefully revised the paper in light of your comments, and please find our response to the comments made below. We marked the modified part of the manuscript in yellow highlight.

Thank you for considering our revised manuscript!

Point 1:  Abstract

-The sentence in line 27-30: “The conditions of suspension culture …” does not contain essential information and can be omitted, and the culture conditions listed in parentheses can be placed in the previous sentence, line 25: “… culture conditions (initial inoculation amount, carbon source, shaking speed, hormone concentration, culture time),…“

-In line 30: “We found that the lowest (5) contamination rate…”, (5) should be placed after “rate”, and also units of measurement added – 5?

-In line 32: “The highest (88.17%) induction rate…”, (88.17%) should be placed after “rate”

-In line 33: to add “induced” after “callus”: “The callus induced was inoculated onto …”

-(2) /line 33/, (3) /line 35/ and [Conclusion] /line 40/ to be omitted

Response 1: Thank you very much for your suggestion. We read the manuscript carefully and accepted your suggestions. we have also corrected the errors in the citations accordingly, as detailed in the corresponding places in the text.

Point 2: Introduction

-I suggest adding an introductory sentence in line 54 after "..risk of extinction [7]": “Plant tissue culture and cell suspension culture are two techniques that can provide alternative methods for rapid cultivation”

-In line 57: “Zhu [9] et al.”, to be “Zhu et al. [9]” The same inaccuracy was comet further on, on line 307, and in other citation as: Luo [12] et al. /line 294/- to be Luo et al.[12], Glagoleva [14], et al./line 374/ - to be Glagoleva, et al.[14]

Response 2: Thank you very much for your suggestion. We read the manuscript carefully and accepted your suggestions. We have made a corresponding addition at line 54 after “. . risk of extinction [7]”, a corresponding addition was made. And we have also corrected the errors in the citations accordingly, as detailed in the corresponding places in the text.

Point 3: Materials and Methods

-Line 377: “Wild P. japonicus aged 2-30 years…”, add “plants”: “Wild P. japonicus plants /aged 2-30 years/…”,

Response 3: Thank you very much for your suggestion. We accepted your suggestions. Additions have been made in the appropriate locations

Point 4: Conclusions

-The first part of this section /lines 514-525/ needs to be edited as it sounds like a result, and my suggestion is to start with a sentence from line 526: This work established a callus induction and suspension culture system for the roots of P .japonicus, including the following steps: disinfection of the root in 75% alcohol for 60 seconds and then treatment in 10% NaClO for 15 minutes; callus induction of the roots in the medium MS + 16.11 μM  NAA + 13.32 μM 6-BA + 30.0 g / L sucrose + 7.8 g / L agar; callus inoculation in MS +  5.37 μM NAA + 13.32 μM 6-BA + 30.0 g / L sucrose + 3.8 g / L gellan gum, subcultured every 21 days; establishment of suspension culture by transferring the callus on growth medium (MS + 5.37 μM NAA + 13.32 μM 6-BA + 30.0 g / L sucrose) and cultured in a shaker at 110 r / min for 21 days; replacing the growth medium with the production medium (MS + 5.37 μM NAA + 13.32 μM 6-BA + 30.0 g / L glucose) for maximum accumulation of total glycosides. The established culture system can promote the accumulation of glycosides in callus cells and provide a basis for large-scale cell culture and industrial production of medicinal glycosides.

Response 4: Thank you very much for your suggestion. We read the manuscript carefully and accepted your suggestions. Please refer to each picture question for details. Refer to lines 513-524 for details carefully and accepted your suggestions. The notation of the measurement unit in this article conforms to the SI system.

Reviewer 3 Report

Comments and Suggestions for Authors

Dear Authors,

I am writing regarding the manuscript with the ID plants-3129180, entitled " Suspension culture and induction of Panax japonicas callus tissue for the production of secondary metabolic active substances".

The topic addressed is interesting, but the manuscript presents some ambiguities.

Firstly, you should clarify a major issue of this manuscript, specifically the determination of glycoside content. The data presented in the manuscript are quite confusing:

(i) The manuscript title suggests that it presents "Suspension culture and induction of Panax japonicas callus tissue for the production of secondary metabolic active substances."

(ii) In the abstract, it states:

Lines 38-40:

"The highest total saponin content in the suspension culture was achieved after replacing the culture medium with the production medium (MS + 5.37 μM NAA + 13.32 μM 6-BA + 30.0 g/L glucose)."

(iii) In Section 2, Results, the authors present some results regarding secondary metabolites in callus, specifically:

Line 153: Figure 4. Comparison of proliferation coefficient and total glycoside content of different cell lines of Panax japonicus.

Line 235: Section 2.4. presents Effects of suspension culture conditions on the content of total glycosides in cells of Panax japonicas.

(iv) In the Materials and Methods section, the method for the quantitative analysis of glycosides is not presented. The authors have an odd chapter for describing these methods:

Lines 508-512:

4.10.4. Detection of total glycoside content in suspension cultured cells

As in 1.2.2.3 Determination method of total glycosides, the content of total glycosides in the culture was determined under different culture conditions. The effects of the different culture conditions on the accumulation of glycosides in P. japonicus were analyzed and compared.

"Total saponins" refers specifically to the saponins in the samples, while "total glycosides" refers to all types of glycosides present, including saponins. You need to make a clear distinction between these terms to avoid confusion. What did you analyze?

Furthermore, the manuscript references additional figures and tables that are not available on the platform.

Author Response

Response to Reviewer 3 Comments

Dear Reviewer,

We are very grateful to you for taking the time to read and modify our article again. We find that your comments play a very important role in improving the quality of our papers. We have carefully revised the paper in light of your comments, and please find our response to the comments made below. We marked the modified part of the manuscript in green highlight

Thank you for considering our revised manuscript!

Point:Firstly, you should clarify a major issue of this manuscript, specifically the determination of glycoside content. The data presented in the manuscript are quite confusing:

(i) The manuscript title suggests that it presents "Suspension culture and induction of Panax japonicas callus tissue for the production of secondary metabolic active substances."

(ii) In the abstract, it states:

Lines 38-40:

"The highest total saponin content in the suspension culture was achieved after replacing the culture medium with the production medium (MS + 5.37 μM NAA + 13.32 μM 6-BA + 30.0 g/L glucose)."

(iii) In Section 2, Results, the authors present some results regarding secondary metabolites in callus, specifically:

Line 153: Figure 4. Comparison of proliferation coefficient and total glycoside content of different cell lines of Panax japonicus.

Line 235: Section 2.4. presents Effects of suspension culture conditions on the content of total glycosides in cells of Panax japonicas.

(iv) In the Materials and Methods section, the method for the quantitative analysis of glycosides is not presented. The authors have an odd chapter for describing these methods:

Lines 508-512:

4.10.4. Detection of total glycoside content in suspension cultured cells

As in 1.2.2.3 Determination method of total glycosides, the content of total glycosides in the culture was determined under different culture conditions. The effects of the different culture conditions on the accumulation of glycosides in P. japonicus were analyzed and compared.

"Total saponins" refers specifically to the saponins in the samples, while "total glycosides" refers to all types of glycosides present, including saponins. You need to make a clear distinction between these terms to avoid confusion. What did you analyze?

Furthermore, the manuscript references additional figures and tables that are not available on the platform.

Response: Thank you very much for your suggestion. We read the manuscript carefully and accepted your suggestions. We analyzed the content as total saponins, and it is true that we confused the concepts of saponins as well as glycosides in our article, which has been revised in the corresponding part of the article. For the Materials and Methods section, the analytical methods for total saponins have been supplemented as detailed in 4.4.

Tables and figures that do not appear on the platform as they appear in the text have been revised and are detailed in Tab. 1 and Supplementary Materials (Fig. S2).

Round 2

Reviewer 3 Report

Comments and Suggestions for Authors

Dear Authors,

Manuscript has been improved.

Line 375 – “ plants aged 2-30 years” ? or 2-3 years

Author Response

Dear Reviewer,

We are very grateful to you for taking the time to read and modify our article again. We have carefully examined the paper in light of your comment, and please find our response to the comment made below.

Point 1: Line 375 – “ plants aged 2-30 years” ? or 2-3 years

Response 1: “ plants aged 2-30 years” is right. We observed the age by  the number of nodes on the roots and stems of the Panax japonicus. Each section  represents the one year in which P. japonicus grows. We used wild P. japonicus of different ages as experimental materials.